# Incorporation of iPSCs together with TERT-immortalized keratinocytes and fibroblasts into reconstructed human gingiva enhances phenotype of gingival epithelium

Lisa-Lee Brueske[1,2], Sanne Roffel[1,2], Stephanie Beekhuis-Hoekstra[2], Helga E. de Vries[2,3], Susan Gibbs[1,2,4]*

1 Department of Oral Cell Biology, Academic Centre for Dentistry Amsterdam (ACTA), University of Amsterdam and Vrije Universiteit Amsterdam, Amsterdam, The Netherlands, 2 Department of Molecular Cell Biology and Immunology, Amsterdam UMC Location Vrije Universiteit Amsterdam, Amsterdam, The Netherlands, 3 Amsterdam Neuroscience, Amsterdam UMC, Amsterdam, The Netherlands, 4 Amsterdam Movement Sciences, Amsterdam UMC, Amsterdam, The Netherlands

* s.gibbs@amsterdamumc.nl

## Abstract

The oral mucosa plays an important role in maintaining oral and systemic health by protecting the body from harmful environmental stimuli and pathogens. Current reconstructed human gingiva models (RhG) serve as valuable testing platforms for safety and efficacy testing of dental materials, however they lack important phenotypic characteristics typical of the gingival epithelium. We aimed to determine whether incorporating induced pluripotent stem cells (iPSCs) into the hydrogel of a cell-line RhG (reconstructed epithelium on fibroblast-populated-hydrogel) would improve its phenotype. Immortalized human gingival fibroblasts were resuspended with and without iPSCs in collagen-fibrin hydrogels and gingival keratinocytes were seeded on top of the hydrogels to construct RhGs. RhGs were cultured at air-liquid interface for 1, 2, 4 and 6 weeks and extensively characterized by immunohistochemistry. *In situ* hybridization for X and Y chromosomes was conducted to identify female iPSCs and male fibroblasts in the RhGs. iPSC-RhGs showed increased epithelial thickening, rete ridge formation, increased cell proliferation and normalized expression of differentiation markers (keratins, involucrin, loricrin, SKALP/elafin) compared to standard RhGs, resulting in an epithelial phenotype very similar to the native gingiva. An increase in apoptotic cells was detected in iPSC-RhGs after 1 week air-exposed culture, and no iPSCs were detected in the hydrogels after 2 weeks air-exposed culture. The increase in apoptotic iPSCs after 1 week air-exposed culture correlated with an increase in keratinocyte proliferation responsible for the superior phenotype observed at 2 weeks.

**Data availability statement:** All quantification data from image analysis is available on Menedley Data (doi:10.17632/drhx7djbcd.1).

**Funding:** This study was supported by the AES programme of the Dutch Research Council (NWO; project number: 19247; recipient: SG and EV) The funders had no role in study design, data collection and analysis, decision to publish, or preparation of the manuscript.

**Competing interests:** The authors have declared that no competing interests exist.

## Introduction

The oral mucosa lines the inside of the mouth forming a protective barrier and therefore plays an important role in the maintenance of oral and systemic health as it protects the body from potentially harmful environmental stimuli and invaders, such as pathogens and leachables from dental materials [1]. The oral mucosa surrounding the teeth is called gingiva and consists of two layers, the epithelium and the underlying lamina propria (LP). The epithelium consists of several layers of densely packed cells which are predominantly keratinocytes with no extracellular matrix, and forms the outermost barrier to the outside environment. The epithelium can be further divided into sublayers, namely the stratum basale, stratum spinosum, stratum granulosum and the stratum corneum [2]. The stratum basale is the first layer of the epithelium which consists of columnar keratinocytes and is connected to the LP via the basement membrane. This layer is characterized by deep invaginations, called rete ridges, which reach into the underlying LP and also by a higher number of proliferating cells which are responsible for the increased thickness of the gingival epithelium. Further towards the apical side of the epithelium is the stratum spinosum, which constitutes the majority of the epithelium containing round keratinocytes. The stratum granulosum consists of progressively flattening keratinocytes and acts as an intermediate layer between the stratum spinosum and the stratum corneum. Finally, the outermost layer of the epithelium is the stratum corneum which consists of dead flattened keratinocytes, which are shed into the oral cavity [2]. The LP is a layer of connective tissue interspersed with fibroblasts which account for the majority of cells within the LP, but it also contains vasculature and several types of immune cells.

Preclinical testing of health care products, drugs and medical devices (*e.g.,* dental implants) which come into contact with the oral mucosa, still heavily relies on the use of animals to assess safety and efficacy before proceeding to testing with human participants [3,4]. However, the translatability of findings derived from animal models to humans is limited due to species-related differences affecting, among other things, immune responses, metabolism and anatomy. Furthermore, due to the high costs and the time required there is little flexibility with regards to experimental design and scalability when using animal models [5]. To overcome these limitations and the ethical concerns of animal testing, physiologically relevant human models are required, which is further emphasized by frameworks such as the 3Rs (Replacement, Reduction, and Refinement) and the FDA Modernization Act 2.0 which encourage the adoption of alternative, more humane research methods. This has boosted the development of microphysiological systems to fill the gap between preclinical testing and testing in humans.

Microphysiological systems aim at culturing cells in a physiologically relevant context. These systems include 3D cell culture models, which can additionally be cultured on organ-on-a-chip platforms which implement microfluidics to subject the cells to fluid flow mimicking the dynamic environment of the tissue in question. Such 3D cell culture models of barrier tissues such as skin, lung or mucosa are either reconstructed models or organoids [6,7]. Typically, these reconstructed models consist of a polarized epithelium with or without a fibroblast-populated hydrogel

(acting as a LP) grown on a porous membrane within a transwell, the cells being derived from cell lines or primary cells [8]. In contrast, organoids are clusters of cells (often spheroid-like) representing the phenotype and cell diversity of a mini-organ and are often generated from pluripotent cells [9]. Pluripotent cells can be differentiated to model the embryonic development of the desired organ. A popular cell source for organoid generation is induced pluripotent stem cells (iPSCs). iPSCs were developed by Yamanaka et al. [10] in 2006 and are generated by reprogramming adult somatic cells into a pluripotent state, thereby regaining their capacity to differentiate into almost any cell type. An advantage of iPSCs is that they can be used to generate patient-specific organoids opening up a lot of possibilities for the field of personalized medicine. Apart from their potential to generate a multitude of different organoids, iPSCs have also been shown to have great regenerative capacities by stimulating neighboring fibroblasts and skin cells [11,12]. For future microphysiological systems it would be advantageous to combine the barrier properties of the reconstructed model with the regenerative properties of iPSCs.

In the past we have developed a reconstructed human gingiva model (RhG) consisting of gingival keratinocytes and gingival fibroblasts entirely from TERT-immortalized cells or primary gingival cells [13]. The gingival keratinocytes ultimately form a stratified, differentiated gingival epithelium on top of the fibroblast-populated collagen hydrogel, once lifted to the air-liquid interface and cultured air-exposed (AE) with culture medium on the basal side only. Histologically, this RhG resembles the native human gingiva in many ways and has been used for extensive research studying oral host-microbiome interactions, wound healing properties as well as the exposure of the oral mucosa to chemical sensitizers or metal ions [14–16]. Similar models have been developed and used by other researchers. While our model incorporates a collagen hydrogel as the LP scaffold, this can be replaced by acellular dermis, a synthetic scaffold, or a gelatin or fibrin hydrogel [17]. Another important variant between the different reconstructed models is the source of the keratinocytes and fibroblasts [17]. While we mainly use TERT immortalized cell lines, also primary cells or other immortalized cell lines have been used. The advantage of immortalized cell lines is that they can be readily amplified and used at a higher passage number than primary cells [17]. Regardless of the scaffold and cell source of choice, these models look histologically quite similar with a differentiated, stratified epithelium consisting of a few layers of keratinocytes on top of the fibroblast populated LP component. While such three dimensional models (ours and models of others) recapitulate the human gingiva more closely than conventional two dimensional cell cultures grown in culture dishes, current RhG models still lack certain features characteristic of native human gingiva [18]. For example they still have a relatively thin epithelium, less keratinocyte proliferation and lack rete ridges. These characteristics are key to the gingiva phenotype distinguishing it from similar barrier tissues such as the skin and contribute to its function, e.g., barrier property and resistance to mechanical loads generated by eating [19].

The aim of this study was to determine whether incorporating iPSCs into the hydrogel of the current cell line RhG would improve its phenotype. Compared to the standard RhG without iPSCs, models containing a mix of fibroblasts and iPSCs in the hydrogel (iPSC-RhGs) showed an increased proliferation of the keratinocytes within the epithelium, increased epithelial thickness, improved differentiation and even invagination of the epithelium into the collagen hydrogel, thus resulting in an epithelial phenotype much more resembling that of the native gingiva.

## Methods

### Cell culture and construction of RhG

**hTERT Keratinocytes:** For the culture of immortalized gingival keratinocytes (hTERT KCs) (KC-TERT, OKG4/bmi1/TERT, Rheinwald laboratory, Boston, MA, USA) Derma life medium (LifeLine Cell Technology, San Diego, CA) and KCI in a 9:1 ratio was used. KCI consisted of a 3:1 mixture of DMEM (Gibco, Grand Island, USA) and Ham's F-12 Nutrient Mix (Gibco, Grand Island, USA) supplemented with 1% penicillin–streptomycin (Gibco, Grand Island, USA), 5% Fetal Clone III (FCIII) (Fisher scientific, Logan, UT, USA), 0.1 μm insulin (Sigma-Aldrich, St. Louis, MO, USA), 1 μm hydrocortisone (Sigma-Aldrich, St. Louis, MO, USA), 1 μm isoproterenol (Sigma-Aldrich, St. Louis, MO, USA) Dermalife – KCI (1:9) was

further supplemented with 2.5 ng/ml epidermal growth factor (EGF) (Sigma-Aldrich, St. Louis, MO, USA). When hTERT KCs reached 80–90% confluency, they were passaged and used for RhG construction.

**hTERT Fibroblasts:** Immortalized gingival fibroblasts (hTERT Fibs; male donor) (Fib-TERT, T0026, ABM, Richmond, BC, Canada) were cultured in DMEM supplemented with 1% penicillin–streptomycin, and 5% FCIII. When 80–90% confluent, hTERT Fibs were passaged and used for RhG construction.

**iPSCs:** iPSCs (episomal hiPSC line, A18945, Gibco, Grand Island, USA; female donor) were cultured in mTeSR™ Plus medium (Stem cell technologies, Vancouver, Canada) on plates coated with human recombinant Laminin521 (BioLamina AB, Sundbyberg, Sweden). Laminin521 is an animal-free product. When 80–90% confluent, iPSCs were passaged as single cells and used for iPSC-RhG construction.

**RhG:** The standard RhG was constructed as previously described [13]. In short, fibroblasts were trypsinized with 0.05% trypsin (Gibco, Grand Island, USA) and resuspended in 1:4 FCIII: Hank's Balanced Salt Solution (HBSS) +Ca+Mg (Gibco, Grand Island, USA) to obtain a concentration of $0.8 \times 10^6$ cells/ml. A hydrogel was constructed consisting of rat tail collagen type I and fibrinogen (Diagnostica Stago S.A.S., Asnieres sur Seine, France) in a 1:1 ratio, containing HBSS –Ca –Mg, (Gibco, Grand Island, USA) and gingival hTERT fibroblasts. Then 0.5 U/ml thrombin (Merck KGaA, Darmstadt, Germany) was added to the collagen-fibrinogen fibroblast mixture and 350ul containing approximately 28.000 fibroblasts per hydrogel was pipetted onto 6.5mm transwells with 0.4 μm pore size (Corning Life Sciences, Corning, NY, USA). Hydrogels were left in the incubator to solidify for 2 hours at 37°C and 5% $CO_2$ and then cultured in DMEM supplemented with 1% penicillin–streptomycin and 5% FCIII for 24 hours. After 24 hours 65.000 hTERT KCs were seeded on top of each hydrogel and the RhGs were cultured in KCI medium. After three days, RhGs were lifted to air-liquid interface to allow differentiation of keratinocytes and cultured in KCII medium consisting of DMEM with Ham's F12 (3:1) supplemented with 1% penicillin–streptomycin, 1% FCIII, 0.1 μm insulin, 2 μm hydrocortisone, 1 μm isoproterenol, 10 μM carnitine (Sigma-Aldrich, Sigma-Aldrich, St. Louis, MO, USA), 10mM L-serine (Sigma-Aldrich), 0.4mM L-ascorbic acid (Sigma-Aldrich, St. Louis, MO, USA), and 2ng/mL EGF. The RhGs were refreshed twice a week and 24 hours before harvesting the medium was changed to KCII medium without hydrocortisone. RhGs were harvested 1 week, 2 weeks, 4 weeks and 6 weeks after AE culture and processed for further analysis.

**iPSC-RhG:** For construction of the iPSC-RhG, iPSCs were dissociated with TrypLE™ Select (Gibco, Grand Island, USA) and resuspended in mTeSR™ Plus and HBSS+Ca+Mg (1:4) to obtain a concentration of $0.8 \times 10^6$ cells/ml. This iPSCs suspension was mixed with the hTERT Fib suspension described above at a 1:1 ratio obtaining an iPSCs:Fibs suspension ($0.4 \times 10^6$ iPSCs/ml and $0.4 \times 10^6$ Fibs/ml), so that the total number of cells was the same as in the standard RHG which contained $0.8 \times 10^6$ Fibs/ml. This iPSCs-Fibs suspension was then used to incorporate into the collagen-fibrinogen hydrogel and further processed in the same way as the standard RhG. iPSC-RhGs were harvested 1 week, 2 weeks, 4 weeks and 6 weeks after AE culture and processed for further analysis.

## Histology and immunohistochemistry

RhG and iPSC-RhG were harvested, embedded in paraffin and cut into 5 μm tissue sections. Sections were stained with hematoxylin and eosin (H&E) for morphological analysis. For immunohistochemistry (IHC), paraffin sections were deparaffinized and further processed based on the primary antibody in question. For the Ki67 (Clone: Mib1, Agilent, Santa Clara, CA, USA) and Vimentin (V9, Agilent, Santa Clara, CA, USA) staining, slides were subjected to antigen retrieval by gently boiling them for 10 minutes in 0.01 M sodium citrate buffer (pH 6.0) and letting them cool down for at least 1.5 hours. For the K14 (RCK107, Monosan, Uden, Netherlands) staining slides were gently boiled in 10mM Tris/ 1mM EDTA buffer (pH 9.0) and let to cool down for at least 1.5 hours. For the K10 (DE-K10, Progen, Heidelberg, Germany) and Loricrin (Poly19051, Biolegend, San Diego, CA, USA) staining protease digestion with pepsin (Agilent, Santa Clara, CA, USA) was performed for 5–8 minutes at 37°C. For the SKALP (TRAB20, HyCult Biotechnology, Uden, Netherlands) and Involucrin (SY5, Novocastra, Newcastle, U.K.) staining, inhibition of endogenous peroxidase was performed by incubating the slides in 0.3% $H_2O_2$ in methanol for 20 minutes. For the α-SMA (1a4, Agilent, Santa Clara, CA, USA) staining no

pretreatment was necessary. After antigen retrieval, pepsin incubation or blocking of endogenous peroxidase, sections were incubated with primary antibodies in 2% goat serum (Agilent, Santa Clara, CA, USA) in PBS for 1 hour. After incubation with the primary antibody, sections were washed with PBS and incubated with BrightVision plus Poly-HRP-Anti-Mouse/Rabbit IgG (VWR International B.V., Breda, Netherlands) and 3-amino-9-ethylcarbazole (AEC) (Sigma-Aldrich, St. Louis, MO, USA) substrate and finally counterstained with hematoxylin. For imaging a Nikon Eclipse 80i microscope (Nikon, Tokyo, Japan) with the NIS Elements 4.13 software (Nikon, Tokyo, Japan) was used as well as an Olympus VS200 Slide Scanner (Olympus Soft Imaging Solutions GmbH, Münster, Germany).

### Fluorescent X; Y chromosome *in situ* hybridization

RhG and iPSC-RhG were harvested, embedded in paraffin and cut into 10 µm tissue sections. To identify the sex of the cells in the RhG and iPSC-RhGs, a fluorescent *in situ* hybridization of X and Y chromosomes was performed with a commercially available kit containing directly labeled fluorescent probes for X/Y (Vysis CEP X SpectrumOrange/Y SpectrumGreen DNA Probe Kit, Abbott Laboratories, Chicago, IL, USA). After deparaffiniazation sections received an antigen retrieval treatment with pepsin as described above. Then slides were washed and probes were added. Hybridization was performed in a prewarmed humidity chamber of 39°C for at least 20 hours. Next, slides were washed with stringent buffer, air-dried and counterstained with DAPI. Whole slide images were acquired with an Olympus VS200 Slide Scanner (Olympus Soft Imaging Solutions GmbH, Münster, Germany).

### TUNEL assay

RhG and iPSC-RhG were harvested, embedded in paraffin and cut into 5 µm tissue sections. The terminal deoxynucleotidyl transferase dUTP nick-end labelling kit (TUNEL Assay Kit–BrdU-Red ab66110; Abcam, Cambridge, UK) was used to visualize and quantify apoptotic nuclei in the RhG and iPSC-RhG. Slides were treated with Proteinase K solution and further treated as previously described [20]. In short, slides were incubated with a DNA labelling solution containing bromolated deoxyuridine triphosphate nucleotide (BrdU), which was incorporated into DNA strand breaks. Subsequently, slides were incubated with an anti-BrdU-Red antibody and imaged with an Olympus VS200 Slide Scanner (Olympus Soft Imaging Solutions GmbH, Münster, Germany).

### Image quantification

**Length of lamina propria-epithelium junction:** H&E stainings from all time points taken with the Olympus VS200 Slide Scanner were imported into QuPath v0.4.0 [21] and a 4000 µm wide x 2000 µm high annotation was placed over each RhG. Within this annotation a line was drawn along the lamina propria (LP) junction in order to quantify the formation of rete ridges forming at the epithelium-lamina propria hydrogel junction. Results are expressed in mm.

**Ki67 proliferation index:** For quantification of the Ki67 staining, pictures of three different areas per hydrogel taken with a Nikon Eclipse 80i were imported into QuPath. The epithelium was manually annotated and the StarDist extension and the "he_heavy_augment" StarDist model were used to automatically count the number of hematoxylin stained nuclei in the annotated epithelium [22]. Ki67 positive cells were then counted manually and the average number of positive cells per 400 µm epithelium calculated.

**Thickness of epithelium:** To quantify the area of the epithelium on a tissue section, which corresponds to the thickness of the epithelium, the total count of hematoxylin stained nuclei per 400 µm width of epithelium was calculated.

### Assessment of contraction and total cell number within hydrogel

Contraction was determined by measuring the height and diameter of the hydrogel over time.

**Height of hydrogel:** Whole-slide images of the H&E stainings from all time points were used to measure the height of each gel. At the center of each gel a vertical line was drawn from the upper border of the hydrogel to the lower border of the hydrogel.

**Diameter of hydrogel:** Whole-slide images of the H&E stainings from all time points were used to measure the diameter of each gel. Three horizontal lines were drawn at different levels of each gel. The average length of these three lines was considered the diameter of the gel.

**Number of cells in hydrogel:** Two 1000 µm wide x 500 µm high annotations were superimposed on the whole-slide H&E stain images from week 1 and week 2 at the top and bottom of the gels, respectively. For some iPSC-RhG gels 1000 µm wide x 250 µm or 125 µm high annotations were added as their overall height was lower than 1 mm. The number of hematoxylin stained nuclei within the two annotations were counted. To account for differences in gel height, the height of the gel was divided by the height of the annotation and the resulting value was multiplied by the number of nuclei counted within the annotations.

## Quantification of male: Female cells in lamina propria

Whole-slide images of the FISH were imported into QuPath v0.5.1 [21]. For each section the lamina propria was annotated and nuclei were detected with the cell detection command. The "subcellular spot detection" command was used to automate the detection of red and green spots corresponding to X and Y chromosomes respectively. Based on the combination of subcellular spots detected in each nucleus, each cells was assigned a label. Cells in which one green and one red spot was detected were labeled as "Male" corresponding to hTERT Fibs, cells in which two red spots were detected were labeled as "Female" corresponding to iPSCs. Cells which contained 1 green dot were labelled as "1 Y chromosome" and cells which contained one red dot were labelled as "1 X chromosome". All cells in which no subcellular spots were detected, were labelled as "No spots" and all remaining cells were labelled as "Unsure". Cells labelled as "No spots" were excluded from further downstream analysis. For each gel the amount of cells labelled as "Male", "Female", "1 Y chromosome", "1 X chromosome" and "Unsure" were shown as percentage of the total number of cells. This division was required since the thickness of the tissue sections did not include the complete cell nuclei but only a section of it leading to only one chromosome being labeled in some cases.

## Quantification of TUNEL assay

Whole-slide images of the TUNEL staining were imported into QuPath v0.5.1 [21]. For each section the lamina propria was annotated and nuclei were detected using DAPI with the cell detection command. A classifier was trained for detecting TUNEL+ cells among the DAPI stained nuclei, classifying all cells into either negative, TUNEL+ nuclei or TUNEL+ cytoplasm. For each gel the amount of apoptotic cells (sum of Tunel+ nuclei and Tunel+ cytoplasm) was shown as percentage of the total cell count. To detect also apoptotic cells in advanced stages of apoptosis in which the DAPI staining was very faint, a classifier was trained to detect all Tunel+ spots (areas > 2 µm) within each annotation. Based on their location these spots were classified into "spots within a cell" (if coordinates fall within boundaries of previously detected cells) and "spots outside cell". The overall number of Tunel+ spots that don't coincide with already detected cells, was shown for the RhG and the iPSC-RhG.

## Statistical analysis

Statistical analysis was performed in GraphPad Prism 9 (GraphPad Software Inc., La Jolla, San Diego, CA, USA). The experiment consisted of at least 3 independent experimental replicates with intraexperimental duplicates. Thickness of the epithelium, length of the LP-epithelium interface, number of Ki67 positive cells, gel diameter and height, cell count in the gel and the number of apoptotic nuclei and tunel+ spots were assessed for statistical significance between the standard RhG and the iPSC-RhG at each time point using an unpaired t-test. Differences were considered statistically significant with a p-value <0.05.

## Results

### iPSCs enhance phenotype of Reconstructed Human Gingiva

The native gingiva consists of a highly proliferating thick keratinized epithelium attached to the basement membrane via deep rete ridges which extend into the LP (Fig 1A). Since iPSCs are stem cells with regenerative potential, RhGs were constructed with or without the addition of iPSCs into the hydrogel to determine whether they were able to improve the phenotype of the standard RhG model (Fig 1B). Over a time period of 6 weeks AE culture both the standard RhG and the iPSC-RhG model formed a gingival epithelium resembling the native gingival epithelium with the formation of a stratum basale, stratum spinosum, stratum granulosum and a stratum corneum, however epithelial development was accelerated in the iPSC-RhG compared to standard RhG (Fig 1C; compare 2 week cultures in particular). Notably, whereas in the standard RhG model the interface of the epithelium and LP remains flat throughout the culture period, clear rete ridge formation is observed in the iPSC-RhG model at 2 weeks of AE culture (Fig 1C), which is significant when the length of the epithelium-LP interface is quantified (Fig 1D). Furthermore, an increase in thickness of the iPSC-RhG epithelium is observed at 2 weeks as shown in tissue histology and quantification of the number of hematoxylin stained nuclei per 400 μm width of the epithelium (p-value < 0.05) (Fig 1C,E). This increase in number of nuclei per 400 μm width of the epithelium paired with the histological analysis shows that the iPSC-RhG resembles the native gingiva much more closely at 2 weeks than the standard RhG. At 4 and 6 weeks of AE culture the histological difference between the standard RhG and the iPSC-RhG becomes less apparent, but a more distinct stratum corneum remains to be seen in the iPSC-RhG (Fig 1C).

### Increased proliferation occurs prior to increased epithelium thickness in iPSC-RhG

In the native gingiva, proliferative cells which stain positive for Ki67 are located in the basal and lower suprabasal layers of the gingival epithelium, often clustering deep within the rete ridges (Fig 2A). Notably, after 1 week of AE culture, the thin epithelium of the iPSC-RhG showed a 2 fold increase in the number of proliferating Ki67 positive cells compared to the standard RhG. This resulted in Ki67 positive cells being abundantly observed in the basal and lower suprabasal cell layers of the 2 week iPSC-RhG cultures, and notably clustering at the bottom of the rete ridges similar to native gingiva (Fig 2A). In contrast, Ki67 positive cells were restricted to the basal layer of the standard RhG and the overall proliferation was lower and relatively stable over the entire 6 week culture period than in the iPSC-RhG or the native gingiva. At 4 and 6 weeks of AE culture proliferation of iPSC-RhG decreased to values only slightly above that found in the standard RhG (Fig 2B and C).

### Incorporation of iPSCs into Reconstructed Human Gingiva results in *in vivo* like differentiation of gingival epithelium

Since the histology of iPSC-RhG was clearly superior to the standard RhG at 2 weeks, we next looked in detail at the expression of specific gingiva differentiation biomarkers (Fig 3). K14 is expressed in the basal and lower suprabasal layers of the native gingival epithelium. Notably, the iPSC-RhG had a very similar K14 expression profile to the native gingiva in contrast to the standard RhG which showed clear aberrant suprabasal expression (Fig 3A). K10, which is expressed in the upper suprabasal layer of the native gingiva, was expressed in the upper suprabasal layers of both the standard RhG and the iPSC-RhG although the expression profile of the iPSC-RhG was clearly more representative of the native gingiva (Fig 3A).

As the gingival epithelium contains a stratum corneum, the expression of cornified envelope proteins Loricrin, Involucrin and Skalp, which are known markers of terminal differentiation were assessed. SKALP, also known as elafin, is also an antimicrobial defense protein [23]. Similar to the native gingiva, the expression of Loricrin, Involucrin and Skalp in the iPSC-RhG was strongest in the upper layers of the epithelium with decreasing intensity towards the basal layer. In

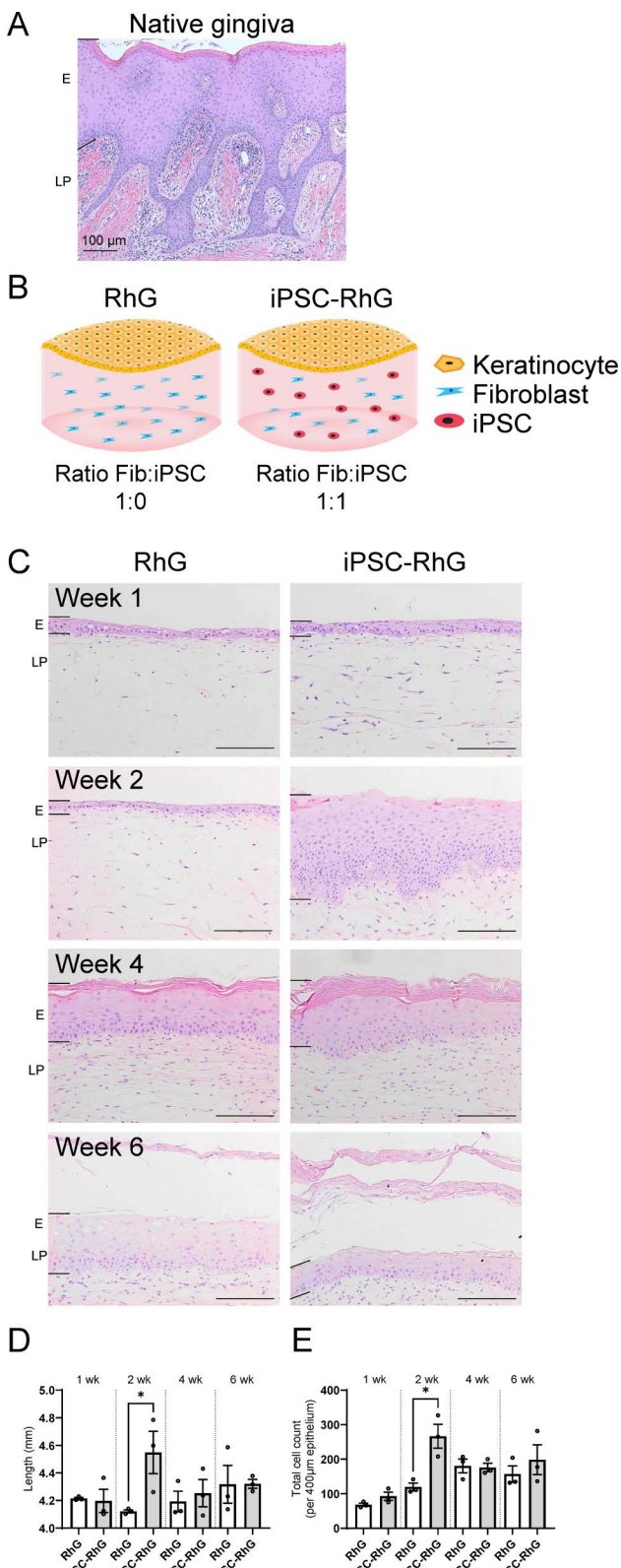

**Fig 1. Histology of native gingiva, experimental design and histological analysis of standard human RhG and iPSC-RhG model.** (A) H&E staining of native gingiva biopsy. (B) Schematic representation of experimental design showing a standard RhG consisting of a collagen hydrogel containing

gingival fibroblasts with gingival keratinocytes on top and iPSC-RhG consisting of a collagen gel containing gingival fibroblasts and iPSCs with gingival keratinocytes on top. (C) Representative pictures of H&E stainings show histology of RhG and iPSC-RhG at 1 week, 2 weeks, 4 weeks and 6 weeks. (D) Quantification of the length of the epithelium-dermis interface. (E) Quantification of the total number of keratinocytes per 400 µm width indicating epithelial thickness. Error bars show SEM of three independent experiments of which the intraexperimental duplicates were first averaged; Non paired t-tests were performed to assess statistical significance between standard RhG and iPSC-RhG at each time point. P-value < 0.05 indicated by asterix. Epithelium (E) and lamina propria (LP) regions are indicated by left hand black bars. Scale bar (bottom right) = 100 µm.

contrast, the expression of these markers was weaker in the standard RhG and limited to a thinner layer in the uppermost epithelium (Fig 3B).

Taken together these results indicate that introduction of iPSCs into the hydrogel of the RhG improves the histology and differentiation of the RhG, as observed by increased epithelial thickening, rete ridge formation, increased cell proliferation and normalized expression of differentiation biomarkers resulting in an epithelial phenotype very similar to the native gingiva. However, the beneficial effect of the iPSCs appears to be restricted to the 2 week AE culture period.

### Presence of iPSCs does not influence total mesenchymal cell number in the hydrogel but does accelerate hydrogel shrinkage

After having characterized the gingival epithelium, the LP hydrogel was assessed to determine the influence of iPSCs. To visualize both the fibroblasts and potential iPSC-derived mesenchymal cells in the hydrogel, tissue sections were immunostained with the mesenchymal cell biomarker Vimentin (Fig 4). In the 2 week standard RhG, Vimentin expressing cells were observed diffusely spread throughout the hydrogel, while cells appeared more compact and densely distributed in the iPSC-RhG (Fig 4A). Next the height and diameter of the hydrogels was quantified in order to determine whether hydrogel shrinkage could be responsible for this observation. The height and diameter of both the standard RhG and the iPSC-RhG hydrogels was similar at 1 week, but decreased significantly in the iPSC-RhG compared to the standard RhG at 2 weeks AE culture. Since the standard RhG also started to shrink at 4 weeks, the differences became less pronounced and by 6 weeks AE culture, both height and diameter of the iPSC-RhG and the standard RhG had decreased to a similar extent. Notably there was no significant difference between the total number of cells present in the hydrogel of the iPSC-RhG and the standard RhG at both 1 and 2 weeks of culture, and even a trend towards lower cell numbers in the iPSC-RhG at 2 weeks culture (Fig 4B).

### Lamina propria contains predominantly gingival fibroblasts after 2 weeks of air exposed culture

As the gingival fibroblasts are male and the iPSCs are female, determining the sex of the cells in the lamina propria hydrogel can elucidate the fate of the iPSCs. Therefore a FISH was performed to stain for the sex chromosomes X (pink stain) and Y (green stain) of cells after 2 weeks of AE culture (Fig 5). In the standard RhG and the iPSC-RhG respectivley an average of 54.7% and 42.9% of the cells present in the lamina propria were clearly identified as male by having one X and one Y chromosome. Only the Y chromosome was detected in 20.1% of the standard RhG and 21.3% of the iPSC-RhG indicating that these cells were also male as the Y chromosome is exclusivley present in male cells. In a similar proportion of remaining cells (23.6% in standard RhG; 33.6% in iPSC-RhG) only a single X chromosome was labelled which did not allow for a definite identification of the cell sex as it could still be either male or female. The incomplete staining of 2 chromosomes within a tissue section was due to the thickness of the tissue sections which did not include the complete cell nuclei but only a section of it. In both the standard RhG and the iPSC-RhG 1% and 1.3% of cells were labeled as female, which is probably due to false positives generated by the automated image quantification as there are no female cells in the lamina propria of the standard RhG. Taken together, on average 74.9% and 64.3% of cells in the lamina propria of the standard RhG and iPSC-RhG respectivley could be identified as male and therefore as fibroblasts with the remaining cells being unclassified. Notably, no female cells and therefore no iPSCs were identified in the lamina propria hydrogel.

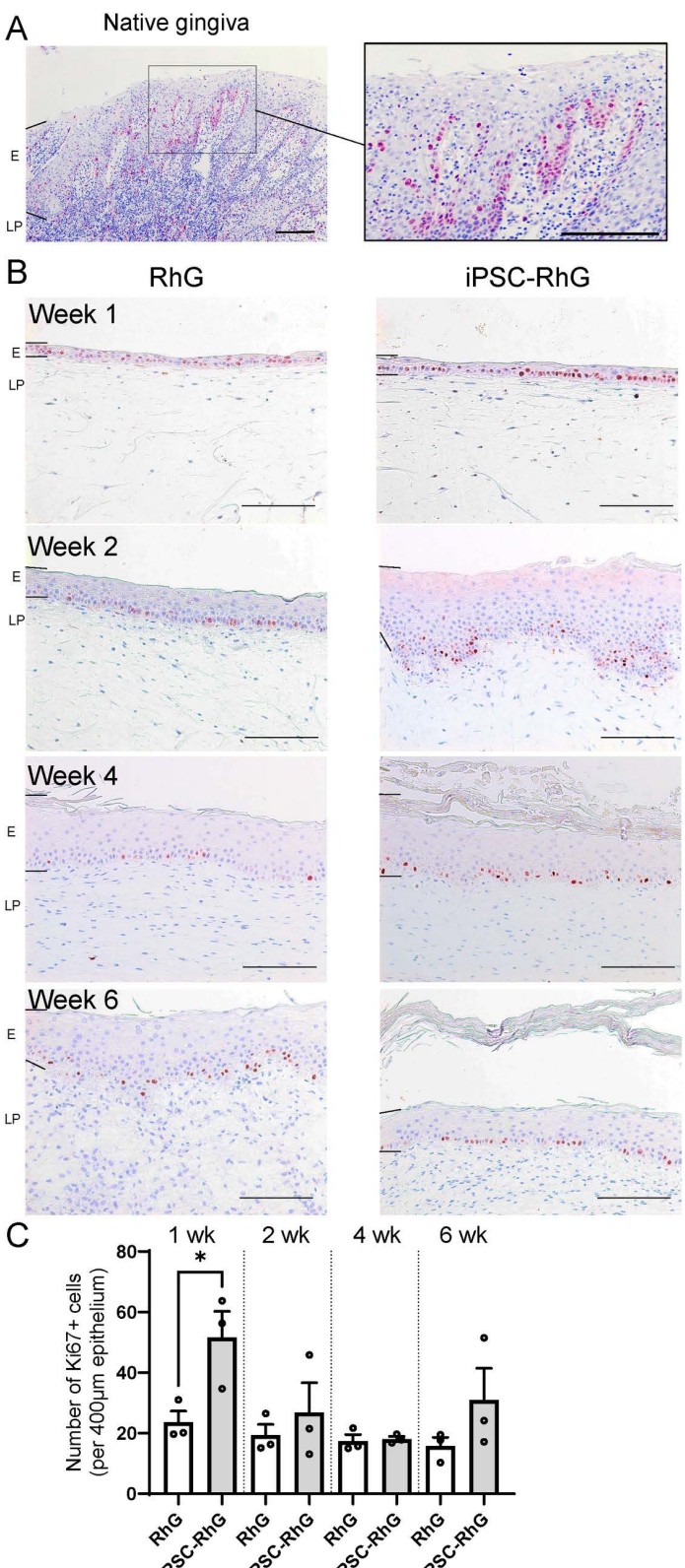

**Fig 2. Comparison of number of proliferating cells in the epithelium of RhG and iPSC-RhG.** Ki67 immunostaining of (A) native gingiva; (B) RhG and iPSC-RhG at 1, 2, 4 and 6 weeks AE culture. Epithelium (E) and lamina propria (LP) regions are indicated by left hand black bars. Scale bar (bottom

right) = 100 µm. (C) Quantification of the number of Ki67 positive cells in the epithelium per 400 µm width. Error bars show SEM of three independent experiments of which the intraexperimental duplicates were first averaged; Non paired t-tests were performed to assess statistical significance between RhG and iPSC-RhG at each time point. P-value<0.05 indicated by asterix.

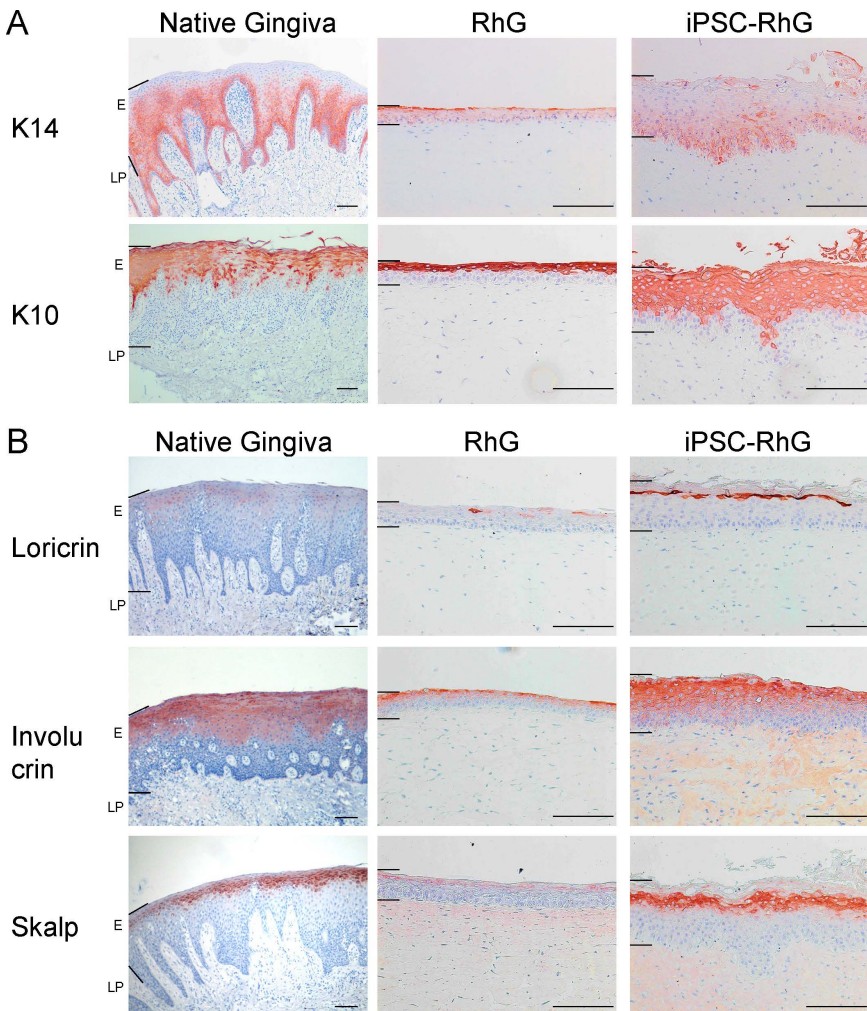

**Fig 3. Immunohistochemical analysis of native gingiva, RhG, and iPSC-RhG.** Representative pictures of (A) keratin expression and (B) cornified envelope precursor protein expression. n=3. Epithelium (E) and lamina propria (LP) regions are indicated by left hand black bars. Scale bar (bottom right) = 100 µm.

## Lamina propria of the iPSC-RhG contains a higher amount of apoptotic cells then the lamina propria of the RhG

As the female iPSCs were not detectable in the lamina propria at 2 weeks, a TUNEL staining was conducted at 1 week of AE culture to determine whether an increase in apoptotic cells can be observed at this earlier timepoint (Fig 6). Cells were classified into apoptotic nuclei, when being DAPI+ and Tunel+, and into Tunel+ spots, when apoptosis was so advanced that DAPI staining was not detectable anymore or below the detection threshold. In the iPSC-RhG 15.72±6.2% of nuclei in the lamina propria were apoptotic, compared to 2.45±1.5% of nuclei in the lamina propria in the RhG (p-value<0.05).

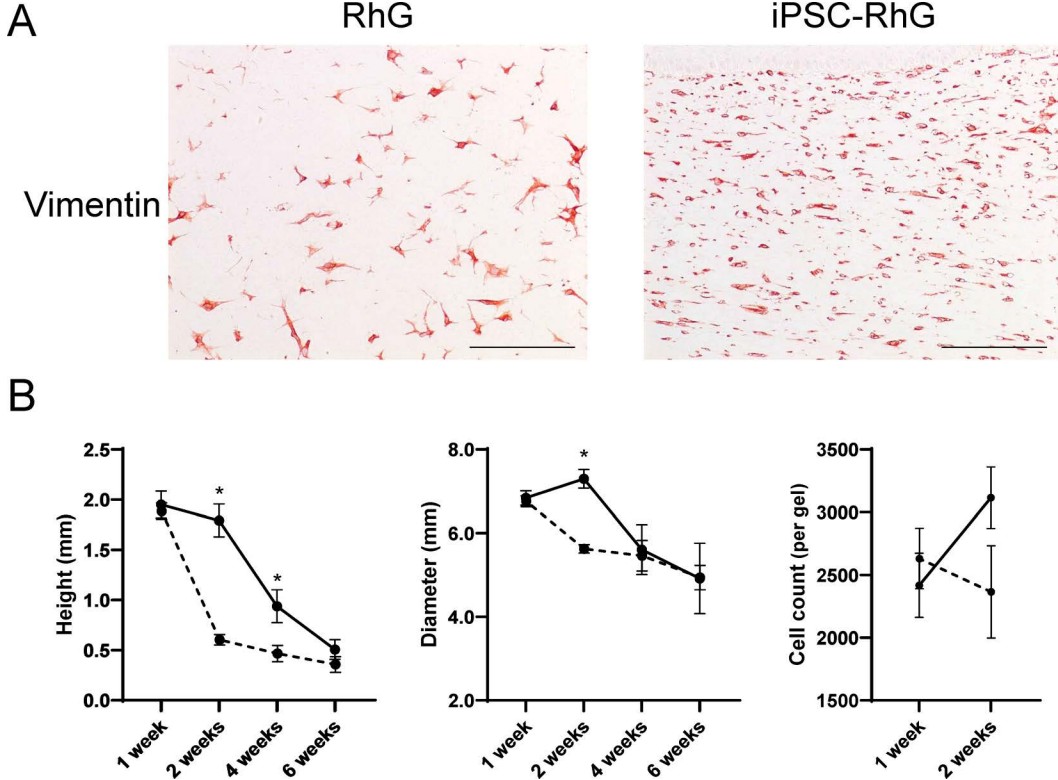

**Fig 4. Characterization of lamina propria of RhG and iPSC-RhG.** (A) Representative pictures of Vimentin staining in RhG and iPSC-RhG shows mesenchymal cell distribution in lamina propria. Scale bar (bottom right) = 100 µm. (B) Quantification of whole-slide images of H&E stainings showing height and diameter of RhG (solid line) and iPSC-RhG (dashed line) gels at 1 week, 2 weeks, 4 weeks and 6 weeks AE culture and cell count per gel in lamina propria at 1 and 2 weeks AE culture. Error bars show SEM of three independent experiments of which the intraexperimental duplicates were first averaged; Non paired t-tests were performed to assess statistical significance between RhG and iPSC-RhG at each time point. P-value < 0.05 indicated by asterix.

In the lamina propria of the iPSC-RhG 102 ± 27 Tunel+ spots were detected compared to 35 ± 17 Tunel+ spots in the lamina propria of the RhG (p-value < 0.05). Taken together these results indicate that the iPSCs became apoptotic in the lamina propria hydrogel at the time when increased epithelial proliferation was observed and prior to epithelial thickening.

## Discussion

In this study we show that the incorporation of undifferentiated iPSCs into a RhG model greatly improves the model's phenotype. This was shown by increased epithelial thickness and rete ridges (invaginations) growing into the underlying LP hydrogel. Notably, this was accompanied by increased keratinocyte proliferation, particularly within the invaginations and an optimal epithelial differentiation marker profile of keratins and cornified envelope precursor proteins after apoptosis of the iPSCs. Taken together, after 2 weeks AE culture, the iPSC-RhG more closely resembled the native gingiva than the standard RhG.

Existing RhGs often have a thin epithelium, and lack rete ridges which are characteristic of the gingival epithelium [24,25]. Rete ridges increase the surface area of the epithelium-lamina propria interface and thereby improve the attachment between the two structures. While little is known about the formation of rete ridges they are believed to form in response to mechanical forces that stimulate the oral mucosa during embryonic development [26–28]. To the best of our knowledge, rete ridges have never spontaneously formed in RhGs or skin models described to date. However, different techniques

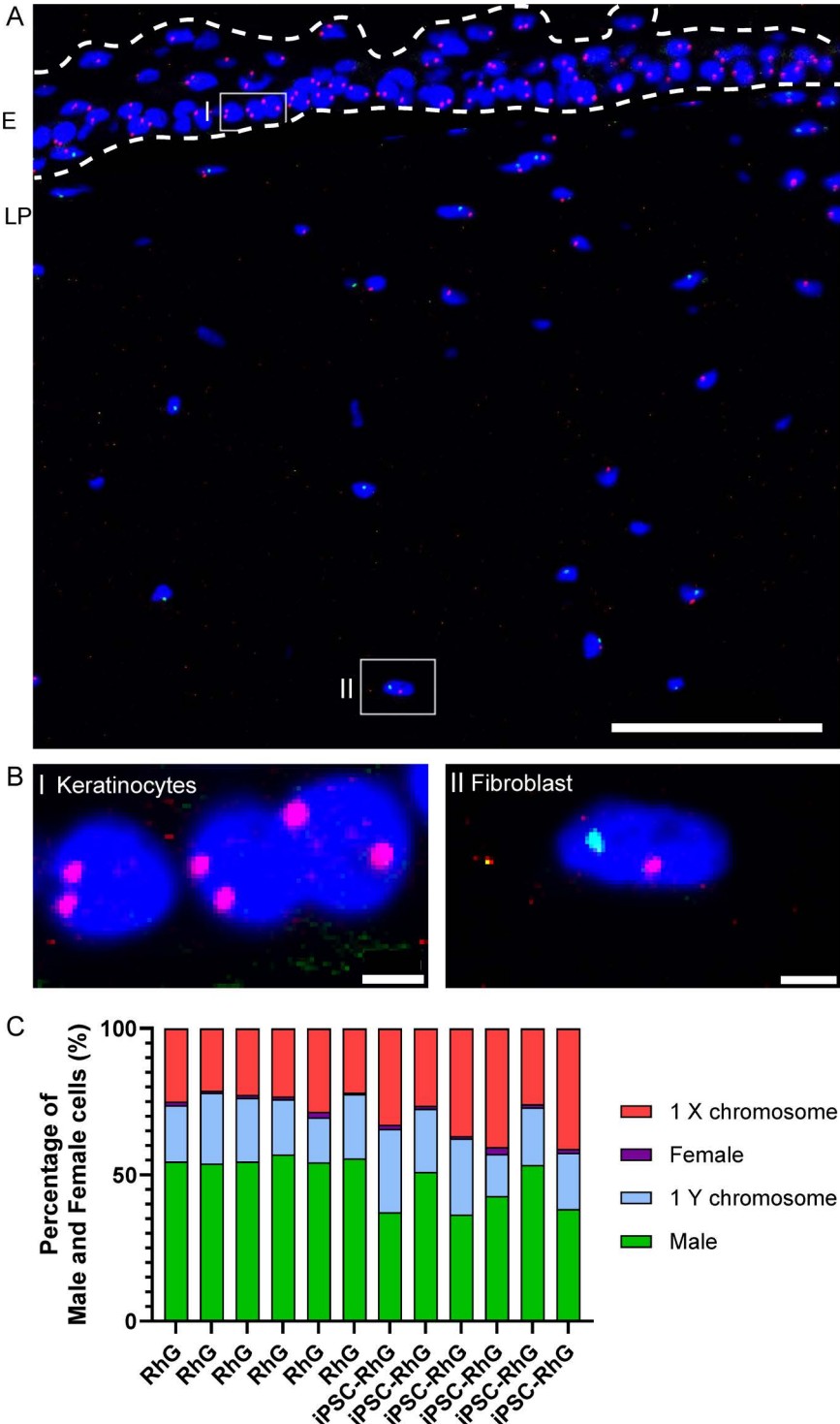

**Fig 5. FISH for X and Y chromosome to identify fate of iPSCs in iPSC-RhG and standard RhG hydrogel after 2 week AE culture.** (A) Representative picture showing X chromosome staining pink; Y chromosome staining green and all nuclei stained with DAPI in dark blue. Epithelium (E) and lamina propria (LP) regions are indicated by white dotted line. Scale bar (bottom right) = 100 µm. (B) Magnified insert of A) showing I) female XX keratinocytes in epithelium and II) male XY fibroblasts in LP hydrogel. Scale bar (bottom right) = 5 µm. (C) Quantification of X/Y chromosome FISH in the LP showing proportion of cells with the following combination of chromosomes: X, XX, Y, XY for each iPSC-RhG and standard RhG at 2 weeks AE culture. n = 3 independent experiments each with an intraexperimental duplicate; each bar represents 1 iPSC-RhG/RhG.

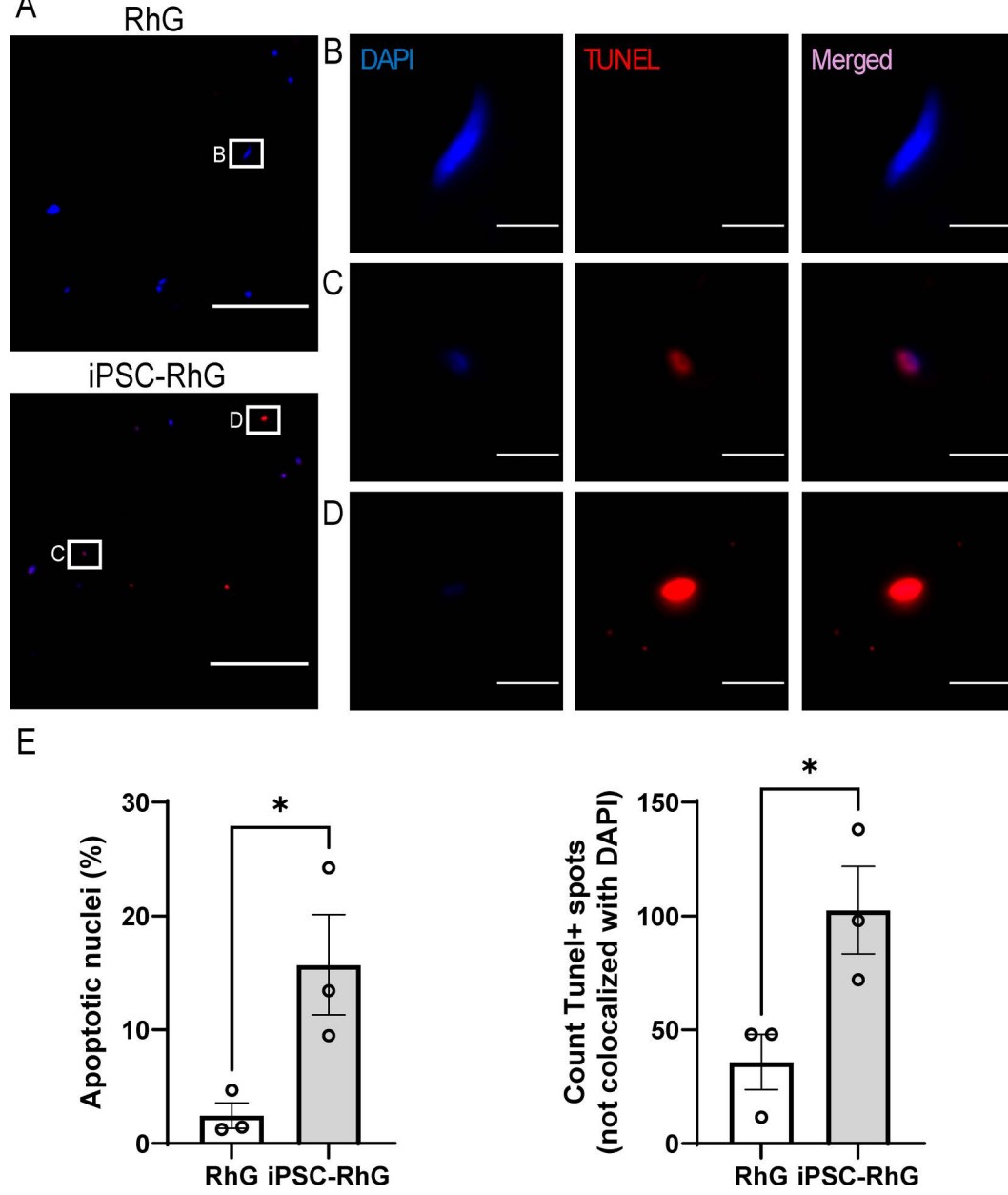

**Fig 6. TUNEL assay of lamina propria of RhG and iPSC-RhG.** (A) Representative pictures of LP of RhG and iPSC-RhG at 1 week AE culture showing nuclei in blue stained with DAPI and Tunel+ nuclei stained in red. Scale bar = 100 µm. (B-D) Magnified insert of A) showing: B) Tunel- nucleus in lamina propria of RhG; C) Tunel+ DAPI+ nucleus in iPSC-RhG; D) nucleus with faint DAPI staining and strong Tunel staining. Scale bar = 10 µm. (E) Quantification of percentage of apoptotic nuclei (left) and number of Tunel positive spots without DAPI staining or DAPI staining below detection threshold (right). Error bars show SEM of three independent experiments of which the intraexperimental duplicates were first averaged; Non paired t-tests were performed to assess statistical significance between standard RhG and iPSC-RhG. P-value < 0.05 indicated by asterix.

have been proposed to incorporate rete ridges into tissue-engineered RhGs or reconstructed human skin models (RhS), for example by micropatterning the scaffold underlying the epithelium with a fractional $CO_2$ laser, by generating micropatterned molds, or by using a de-epidermized dermis [29–32]. While these approaches create structures that histologically resemble

rete ridges, they do not actually model rete ridge formation. Not only do we propose an approach that intrinsically induces the formation of rete ridges by adding iPSCs into our RhG model, our method is also less technically challenging to implement.

The most optimal duration of culture was 2 weeks AE culture for the iPSC-RhG. At 4 and 6 weeks the beneficial effects of adding iPSCs to the hydrogel were much less apparent as the rete ridges flattened out, the epithelium thinned and a thickened stratum corneum formed which sloughed off, with the result that the standard RhG and the iPSC-RhG become histologically more similar. This series of events could reflect the ageing process of the iPSC-RhG, as, even though it has not been shown for gingiva, aged skin shows flattened rete ridges and a thinning of the epithelium contributing to a worsening of the biomechanical properties of the skin [26,33,34]. After 2 weeks AE culture, the iPSC-RhG consists of a clearly defined basal layer, suprabasal layer and stratum corneum, each with corresponding biomarker expression. Even though these markers are expressed in other oral mucosa (RhG) models, they often lack a clear structural separation and tend to be expressed diffusely throughout the epithelium [35–38]. Since, at 2 weeks, the stratum corneum in the iPSC-RhG is thicker than that in the standard RhG, the addition of iPSCs into the hydrogel probably accelerated the terminal differentiation process of the epithelium, which is responsible for the native gingiva-like phenotype of the iPSC-RhG at 2 weeks. For its potential future use as a testing platform in which high throughput is often required in the shortest possible time, an extended culture period is not desirable, and the fact that this improved phenotype can already be achieved after 2 weeks of AE culture is beneficial.

Our observation that the difference between the standard RhG and the iPSC-RhG decreased over time indicates that the iPSCs transiently effected the surrounding cells and are thus responsible for the improved phenotype of the iPSC-RhG. At 1 week, the iPSCs stimulated the proliferation of the gingival keratinocytes, which then drove the thickening of the epithelium and subsequent invaginations into the hydrogel. Several researchers have reported that the regenerative properties of iPSC-derived cells are superior to those of cells derived from other sources, such as umbilical cord cells [39,40]. Oh et al. [12] reported that exosomes from undifferentiated iPSCs have beneficial effects on human dermal fibroblast, such as stimulating fibroblast proliferation and protecting them from damage caused by environmental factors such as UV radiation. In terms of oral rete ridge formation, Wu et al. [27] reported that the proliferation of oral keratinocytes can be stimulated by mechanical loading resulting in ERK1/2 activation. It is therefore possible that the iPSCs intrinsically stimulate the proliferation of the gingival keratinocytes by triggering similar mechanisms exerted by mechanical forces, such as the activation of the ERK signaling pathway. It cannot be ruled out that the shrinkage observed in the iPSC-RhG between weeks 1 and 2 of AE culture contributed to the phenotypical changes observed in the iPSC-RhG. However, shrinkage of collagen-based hydrogels is a common observation in RhG and RhS cultures and we and others have not found this to coincide with the formation of rete ridges [25,41].

Our past work on skin models showed that increased fibroblast numbers in the hydrogel will influence epidermal proliferation [42]. This led us to examine whether this could also be the case with the iPSC-RhG. A closer look at the lamina propria hydrogel of the standard RhG and the iPSC-RhG suggested a higher density of the mesenchymal cells in the lamina propria of the iPSC-RhG than in the standard RhG; this density correlated with an increased contraction (decreased height and diameter) of the iPSC-RhG. However, the total number of mesenchymal cells throughout the gels was similar in both conditions at 1 week and 2 weeks of AE culture. This means that, the differences observed in the iPSC-RhG epithelium were not due to higher mesenchymal cell numbers in the hydrogel.

The increase in apoptotic cells in the lamina propria of the iPSC-RhG at 1 week AE culture combined with the fact that the female iPSCs were no longer detectable in the lamina propria at 2 weeks AE culture, indicates that the iPSCs die in the lamina propria at an early stage. Bock et al. [43] reported that apoptotic stem cells release higher amounts of FGF2, which leads to a stimulation of keratinocyte proliferation and promotes wound healing in skin. Therefore, it is most likely that that the apoptotic iPSCs secreted factors stimulating their surrounding cells, which leads to the high number of fibroblasts in the hydrogel and the improved phenotype of the iPSC-RhG model. This is supported by the fact that, at 2 weeks AE culture, an equal number of fibroblasts was found in both models, even though half the number of fibroblasts

was seeded into the iPSC-RhG than into the standard RhG. Furthermore, Romana-Souza et al. [44] reported that dermal fibroblasts are capable of phagocytosing apoptotic endothelial cells and that this phagocytotic activity goes in hand with a switch towards more pro-healing phenotype, responsible for increased collagen gel contraction, which was also observed in our study.

Limitations of our study include the use of one commercially available iPSC line. Other iPSC lines would need to be tested to see whether the achieved effect is line-specific or can be reproduced with different lines. Furthermore, in the current study the factors released by the apoptotic iPSCs which are responsible for the observed effect on the epithelium were not identified.

Future research should focus on identifying the factors responsible for the stimulatory effect that apoptotic iPSCs have on the surrounding cells. These factors might then be used to supplement the culture medium of standard RhGs resulting in an improved phenotype. Furthermore, it should be determined whether the iPSC-RhG also has improved functionality compared to the standard RhG by testing its barrier function, *e.g.,* upon exposure to chemicals [45].

In conclusion, the incorporation and subsequent apoptosis of iPSCs into the fibroblast-collagen gel of the gingiva model accelerates the differentiation process of the gingival epithelium leading to an improved phenotype more closely resembling the native gingiva. We show that the regenerative potential of iPSCs triggered by their apoptosis can be used to improve the phenotype of reconstructed organotypic models.

## Acknowledgments

The authors acknowledge the fruitful discussions with Albert Feilzer (ACTA) and the BodyBarriers consortium.

## Author contributions

**Conceptualization:** Lisa-Lee Brueske, Stephanie Beekhuis-Hoekstra, Susan Gibbs.

**Funding acquisition:** Helga E de Vries, Susan Gibbs.

**Investigation:** Lisa-Lee Brueske, Sanne Roffel.

**Methodology:** Lisa-Lee Brueske, Stephanie Beekhuis-Hoekstra, Susan Gibbs.

**Resources:** Stephanie Beekhuis-Hoekstra, Susan Gibbs.

**Supervision:** Helga E de Vries, Susan Gibbs.

**Writing – original draft:** Lisa-Lee Brueske, Susan Gibbs.

**Writing – review & editing:** Stephanie Beekhuis-Hoekstra, Helga E de Vries, Susan Gibbs.

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
