## [Decision Letter · Decision Letter 0]

Dear Dr. Gibbs,

Thank you for submitting your manuscript to PLOS ONE. After careful consideration, we feel that it has merit but does not fully meet PLOS ONE’s publication criteria as it currently stands. Therefore, we invite you to submit a revised version of the manuscript that addresses the points raised during the review process.

We look forward to receiving your revised manuscript.

Kind regards,

Xinjun Lu

Academic Editor

PLOS ONE

Journal Requirements:

Reviewers' comments:

Reviewer's Responses to Questions

**Comments to the Author**

1. Is the manuscript technically sound, and do the data support the conclusions?

Reviewer #1: Yes

Reviewer #2: Yes

2. Has the statistical analysis been performed appropriately and rigorously?

Reviewer #1: Yes

Reviewer #2: Yes

3. Have the authors made all data underlying the findings in their manuscript fully available?

Reviewer #1: Yes

Reviewer #2: Yes

4. Is the manuscript presented in an intelligible fashion and written in standard English?

Reviewer #1: Yes

Reviewer #2: Yes

Reviewer #1: Referring to line 82, mentioning the 3Rs could further enhance the scientific soundness of the statement: https://nc3rs.org.uk/who-we-are/3rs. Additionally, authors could mention the FDA Modernization Act 2.0.

The authors should mention the terms microphysiological system and state organ-on-chip models. (line 86)

Referring to lines 89 to 115, the authors must focus on the main thesis statement and limit the text related to organoids.

Referring to lines 123 to 126, the authors are advised to compare the models they mentioned. It would help the reader better understand the current model's novelty and limitations.

The current research topic is interesting; authors should also consider providing a graphical abstract of their study.

Authors must consistently use the same abbreviation of induced pluripotent stem cells throughout the manuscript.

iPSCs are typically cultured on matrigel-coated surfaces; why do authors prefer Lamaninin over matrigel?

The authors should mention their study's limitations in the discussion section and expand the section on its potential implications.

Authors must label all the histological images with the proper cell type/layer.

Reviewer #2: The study of Brueske et al, aimed to determine whether incorporation of the human iPSCs into the hydrogel of the human TERT-immortalized keratinocytes and fibroblasts would improve the phenotype of the reconstructed gingival epithelium. Interestingly that such reconstructed human gingiva models demonstrated augmented keratinocyte proliferation within the epithelium, enhanced the thickness of the resultant epithelium, improved differentiation and interdigitation of the epithelium into the collagen hydrogel compared to the standard model developed from TERT cells only. Based on these data the authors conclude that the new iPSC-supplemented gingival model phenotypically resembles the native gingiva much accurately than the earlier models. The study characterises the models with examining morphological changes, i.e.: H&E histology and immunohistochemistry chromosome in situ hybridization and the TUNEL assay. Overall, the data look convincing, however, it is highly recommended that in the future studies the authors will employ combined assessment of structural and physiological changes, for the bilateral characterisation of the models and matching of the form and function.

There are few aspects after considering of which the manuscript could be accepted for publication:

• The images with the histology are of very low quality. Dear authors, please provide the images of a publication quality.

• The figure legends are provided inside the chapters of the Results section, which is a bit unusual and inconvenient. Dear authors, please place the figure legends separately, after the References, or after the Figures.

• In the Figures the P-values are represented as < 0.05, whereas in the text the p-values are given as a certain number, e.g. = 0.0424. Dear authors, please make it uniform for both.

• In the Figure 6 the images A-D contain the value of the scale bars (i.e. 10um, 100um), whereas the other Figures (5 B, C), 4 (A), 3 (A, B), 2 (A, B), and 1 (A, C) do not have these values. Dear authors, please make things uniform.

• In the experiments with the lamina propria (Figure 6) the number of the experiments is not stated. Dear authors, please provide the n-numbers.

• The Latin used throughout the text must be Italic, i.e.: in situ – line 43; e.g. – lines 76, 102, 105, 128; in vitro – line 357, etc.

**Do you want your identity to be public for this peer review?** For information about this choice, including consent withdrawal, please see our Privacy Policy

Reviewer #1: No

Reviewer #2: No

---

## [Author Response · Author response to Decision Letter 1]

5 Jun 2025

June 05, 2025

Response Letter Research Article, Manuscript Number: PONE-D-25-15760

Dear Editor,

We would like to thank you and the reviewers for providing valuable comments and for their time and effort reviewing our work. These reviewer comments combined have provided constructive and helpful criticism of our work and allowed us to further improve the quality of our manuscript.

We have clearly identified the responses in the letter and kept the track changes on in the manuscript. As requested we also provide an unmarked version of the revised manuscript. Please find the response to reviewer questions below and revised manuscript.

We appreciate your consideration of our manuscript for publication in PLOS ONE.

With Kind Regards,

Sue Gibbs

Answer: The style requirements have been checked and the manuscript meets the style requirements.

Answer: All authors agree with making data available on Mendeley Data. The data has been placed in a repository, which will be made public upon acceptance of our manuscript.

Answer: The corresponding author ORCID ID has been added.

Answer: We have removed the section referring to the data not shown, as it is no a core part of the research presented. See revised text: Page 18, line 429-433.

Answer: The reference list has been reviewed for completeness and correctness. No retracted papers are included in the reference list.

Review Comments to the Author

Reviewer #1

Comment 1:

Referring to line 82, mentioning the 3Rs could further enhance the scientific soundness of the statement: https://nc3rs.org.uk/who-we-are/3rs. Additionally, authors could mention the FDA Modernization Act 2.0.

Answer: See revised introduction: Page 4, line 85-89, “To overcome these limitations …”.

Comment 2:

The authors should mention the terms microphysiological system and state organ-on-chip models. (line 86)

Answer: See revised introduction: Page 4, line 90-96, “This has boosted …”.

Comment 3:

Referring to lines 89 to 115, the authors must focus on the main thesis statement and limit the text related to organoids.

Answer: See revised introduction: Page 5.

Comment 4:

Referring to lines 123 to 126, the authors are advised to compare the models they mentioned. It would help the reader better understand the current model's novelty and limitations.

Answer: See revised introduction: Page 6, line 136-145, “Similar models have been developed …”.

Comment 5:

The current research topic is interesting; authors should also consider providing a graphical abstract of their study.

Answer: Thank you for the suggestion. We have added a graphical abstract.

Comment 6:

Authors must consistently use the same abbreviation of induced pluripotent stem cells throughout the manuscript.

Answer: The manuscript has been revised and the abbreviations have been updated.

Comment 7:

iPSCs are typically cultured on matrigel-coated surfaces; why do authors prefer Lamaninin over matrigel?

Answer: The justification for the use of laminin as coating solution has been added to the subsection “iPSCs” of the Materials and Methods section. See revised materials & methods section: Page 8, line 180-181, “Laminin 521 is an animal-free product.”.

Comment 8:

The authors should mention their study's limitations in the discussion section and expand the section on its potential implications.

Answer: See revised discussion: Page 24, line 581-585, “Limitations of this study include …”.

Comment 9:

Authors must label all the histological images with the proper cell type/layer.

Answer: The figures and legends have been revised. We added labels for the epithelium and lamina propria to all figures containing histological images, as well as labels for cell type to the figure 5 to make it easier to distinguish between fibroblasts and keratinocytes.

Reviewer #2

Comment 1:

The study of Brueske et al, aimed to determine whether incorporation of the human iPSCs into the hydrogel of the human TERT-immortalized keratinocytes and fibroblasts would improve the phenotype of the reconstructed gingival epithelium. Interestingly that such reconstructed human gingiva models demonstrated augmented keratinocyte proliferation within the epithelium, enhanced the thickness of the resultant epithelium, improved differentiation and interdigitation of the epithelium into the collagen hydrogel compared to the standard model developed from TERT cells only. Based on these data the authors conclude that the new iPSC-supplemented gingival model phenotypically resembles the native gingiva much accurately than the earlier models. The study characterises the models with examining morphological changes, i.e.: H&E histology and immunohistochemistry chromosome in situ hybridization and the TUNEL assay. Overall, the data look convincing, however, it is highly recommended that in the future studies the authors will employ combined assessment of structural and physiological changes, for the bilateral characterisation of the models and matching of the form and function.

Answer: We thank the reviewer for these suggestions and agree that future work should look into functionality of the model enhanced with the iPSCs, as well as look into the mechanisms underlying the effect the iPSCs have on the surrounding cells. This has now been added as a future perspective.

See revised discussion: Page 25, line 589-591, “Furthermore, it should be determined…”.

Comment 2:

The images with the histology are of very low quality. Dear authors, please provide the images of a publication quality.

Answer: In the PDF build for submission the figures are shown in lower resolution and the higher resolution images can be accessed by clicking on the link in the upper corner. We have also updated the figures with even higher resolution images.

Comment 3:

The figure legends are provided inside the chapters of the Results section, which is a bit unusual and inconvenient. Dear authors, please place the figure legends separately, after the References, or after the Figures.

Answer: We are following the guidelines for authors from PLOS One, which request the figure captions to be placed in the manuscript text in read order, immediately following the paragraph where the figure is cited.

Comment 4:

In the Figures the P-values are represented as < 0.05, whereas in the text the p-values are given as a certain number, e.g. = 0.0424. Dear authors, please make it uniform for both.

Answer: The manuscript has been updated and the p-values are now mentioned as < 0.05.

Comment 5:

In the Figure 6 the images A-D contain the value of the scale bars (i.e. 10um, 100um), whereas the other Figures (5 B, C), 4 (A), 3 (A, B), 2 (A, B), and 1 (A, C) do not have these values. Dear authors, please make things uniform.

Answer: All figures have been updated and the values of the scale bars are now exclusively in the legend of the figures.

Comment 6:

In the experiments with the lamina propria (Figure 6) the number of the experiments is not stated. Dear authors, please provide the n-numbers.

Answer: The figured legend has been revised and the number of experiments is now stated. See revised legend: Page 21, line 494-496.

Comment 7:

The Latin used throughout the text must be Italic, i.e.: in situ – line 43; e.g. – lines 76, 102, 105, 128; in vitro – line 357, etc.

Answer: The manuscript has been updated and the corresponding text changed to italics.

---

## [Decision Letter · Decision Letter 1]

Incorporation of iPSCs together with TERT-immortalized keratinocytes and fibroblasts into reconstructed human gingiva enhances phenotype of gingival epithelium.

PONE-D-25-15760R1

Dear Dr. Gibbs,

We’re pleased to inform you that your manuscript has been judged scientifically suitable for publication and will be formally accepted for publication once it meets all outstanding technical requirements.

Kind regards,

Xinjun Lu

Academic Editor

PLOS ONE

Additional Editor Comments (optional):

Reviewers' comments:

Reviewer's Responses to Questions

**Comments to the Author**

Reviewer #1: All comments have been addressed

Reviewer #2: All comments have been addressed

2. Is the manuscript technically sound, and do the data support the conclusions?

Reviewer #1: Yes

Reviewer #2: Yes

3. Has the statistical analysis been performed appropriately and rigorously?

Reviewer #1: Yes

Reviewer #2: Yes

4. Have the authors made all data underlying the findings in their manuscript fully available?

Reviewer #1: Yes

Reviewer #2: Yes

5. Is the manuscript presented in an intelligible fashion and written in standard English?

Reviewer #1: Yes

Reviewer #2: Yes

Reviewer #1: The authors revised the manuscript according to the recommendations and provided a satisfactory response to the queries raised. Considering this, the manuscript appears suitable for publication in PLOS One.

Reviewer #2: I am satisfied with the revisions made to the manuscript by Brueske et al. and have no objections to its acceptance for publication in PLOS ONE.

**Do you want your identity to be public for this peer review?** For information about this choice, including consent withdrawal, please see our Privacy Policy

Reviewer #1: No

Reviewer #2: **Yes: ** Dr Vsevolod Telezhkin

---

## [Editor Report · Acceptance letter]

PONE-D-25-15760R1

PLOS ONE

Dear Dr. Gibbs,

I'm pleased to inform you that your manuscript has been deemed suitable for publication in PLOS ONE. Congratulations! Your manuscript is now being handed over to our production team.

Kind regards,

on behalf of

Dr. Xinjun Lu

Academic Editor

PLOS ONE